# Contribution to Rail System Revitalization, Development, and Integration Projects Evaluation: A Case Study of the Zadar Urban Area

**Maja Ahac** *[ID], **Saša Ahac** [ID], **Igor Majstorović** [ID] and **Željko Stepan**

Department of Transportation Engineering, Faculty of Civil Engineering, University of Zagreb, Fra Andrije Kačića-Miošića 26, 10000 Zagreb, Croatia; sasa.ahac@grad.unizg.hr (S.A.); igor.majstorovic@grad.unizg.hr (I.M.); zeljko.stepan@grad.unizg.hr (Ž.S.)

*   Correspondence: maja.ahac@grad.unizg.hr

**Abstract:** This paper aims to contribute to the process of evaluating urban rail infrastructure projects through the presentation of the methodology and the results of a preliminary feasibility study concerning the revitalization, development, and (re)integration of the rail with road, maritime, and air transportation in the Zadar urban area. The analysis included the identification and evaluation of rail infrastructure alignment variants that would ensure the revitalization of the existing railway infrastructure, relocation of freight rail traffic from the narrow and densely developed suburban coastal area, promotion of intermodal passenger and freight transportation, improvement of urban and regional accessibility and connectivity, increase of traffic safety, reduction of travel time and operating costs, and decrease of traffic impacts on the environment. By consulting legal frameworks, spatial planning documentation, and analyzing the socio-economic context and existing transportation infrastructure function, six variants for the (re)development of the rail infrastructure were designed. As their design approached the area's transportation issues from different angles and could contribute differently to the area's economic, social, and territorial issues, a multi-criteria analysis supplemented with a partial cost–benefit analysis was conducted to select the most suitable variant. The evaluation was based on seven weighted criteria quantified by the normalization of 32 indicator values, scored from 1 to 5, where a score of 5 was considered the highest. Weighting the scores according to the ratios determined through a consultation process with stakeholders resulted in ranking the best variant with a total score of 3.7 and the worst one with a total score of 2.6. To avoid potential objections that the set of criteria weights used was subjective and the result biased, a sensitivity analysis was carried out by systematically varying the weights among criteria. The results showed that the best-ranked variant was also the least sensitive to applied weight shifts, with a score range of 0.2.

**Keywords:** railway infrastructure; urban mobility; multi-criteria analysis; cost–benefit analysis; sensitivity analysis

## 1. Introduction

Urban mobility in Europe accounts for 23% of all road transportation $CO_2$ emissions and up to 70% of other pollutants from transportation. The question of how to enhance urban mobility while, at the same time, reducing congestion, accidents, and pollution is a universal challenge for all major cities [1]. The city of Zadar, the fifth largest city in the Republic of Croatia and an important urban, tourist, cultural, and development center of the Adriatic region, also shares these challenges, as the personal car (PC) is the dominant mode of transportation in the Zadar urban area (ZUA) [2].

Numerous strategies at the international, national, regional, and local (city) levels have been adopted to encourage rail travel as a measure to mitigate PC emissions [3]. It is a common conclusion that people have less need to use PCs in urban areas with diverse land use and an efficient and integrated public transportation system and that the integration of

a rail system in urban transportation is the most efficient way to minimize carbon emissions, improve property values and traffic safety, and reduce congestion [4,5]. However, a gradual but constant increase in private mobility (dating back to the second half of the last century) has caused the shutdown of many existing local and/or urban rail lines—rarely used and, therefore, leading to little or even no profit [6]. Such is the case with the rail line for international, mixed traffic that runs through the coastal part of the ZUA and ends at the railway station in the Zadar city center. Namely, due to a lack of investment in maintaining the functional capacity, development, and modernization of the rail system, passenger rail transportation has become completely uncompetitive and was discontinued ten years ago. Rail freight transportation is also uncompetitive due to the lack of a logistics chain connecting different modes of transportation [2].

Disused railways worldwide are viewed as a negative space in cities and are often an urban blight that results in a poor-quality urban environment causing many adverse effects. Perhaps the biggest negative effect is reflected through their impact on urban traffic flows and city development, as they block traffic and divide urban spaces. The renewal of these rail transportation facilities in urban areas has become increasingly important, considering their prime location (close to, or even within, the central districts of cities) and the growing requirements for mobility and connectivity [7].

As finding a new use for disused railways provides an opportunity for low-carbon travel experiences, attracts new public transportation users, and arrests decay processes, disused railway sites are becoming a focus of redevelopment projects in many European cities [8,9]. Three common strategies for reusing disused railways include converting them into new rail transit systems, redeveloping the urban land around them for different purposes, or shaping new public spaces. Different reuse strategies have different positive effects on urban space, including the improvement of traffic conditions and the opportunities for land development [7].

Zadar City's officials recognized that the existing transportation system was limiting the development of the ZUA and that the increase of accessibility, mobility, and competitiveness of this area could be possible by improving the quality of its rail network while ensuring the physical, operational, and organizational integration of all forms of transportation [2]. Investments in the ZUA rail infrastructure require significant financial resources that could be secured through the World Bank, the European Development Bank, or European Union funds. However, the possibility of using these funds must be preceded by a justification of the need for investment through the evaluation of planned projects.

Evaluation is a set of activities aimed at determining the justification and acceptability of a particular project, choosing the optimal variant of infrastructure development, and, at the same time, determining the level of investment priority in a specific project. Not so long ago, the choice of the optimal variant included only the analysis of financial criteria through least cost analysis. The step up was cost-effectiveness analysis, based on the identification of so-called benefit–cost ratios. It led to identifying the "value for money" of variants [10]. On a strategic level, SWOT analysis and context analysis provide a qualitative description of the research subject and, as a rule, represent a valuable instrument for evaluating projects. Such analyses can successfully identify the drivers and barriers to resuming a service on an abandoned railway corridor [11,12]. Today, to choose a solution that enables the sustainable development of transportation infrastructure, it is necessary to take into account a large number of evaluation criteria: (1) environmental, including resource use and residuals production; (2) economic, including expenditures (capital, operation, and maintenance) and investment in innovation; (3) engineering, including performance; (4) social, including accessibility, acceptability, health; and (5) safety [13]. That is why assessments are performed with cost–benefit analysis (CBA) and/or multi-criteria analysis (MCA).

The CBA approach translates Infrastructure project impacts into comparable monetary units, enabling the observation of incremental changes in welfare resulting from the implementation of a project relative to a counterfactual scenario (the economy without the

project), to examine whether society is better off with the intervention [14]. CBA is still widely used in the evaluation of major transportation investment projects [15,16], although it has been found that it has certain limitations when incorporating and assessing criteria such as environmental or social issues [17].

The MCA approach consists of scoring and ranking the proposed infrastructure development variants by weighting predefined criteria imposed on sets of indicators as the measures of infrastructure performance [18]. It provides the possibility of incorporating factors otherwise difficult to quantify or monetize, as it extends the decision-making process beyond the practical reach of CBA and complements the monetary, financial, and economic considerations with a wider range of criteria. Criteria such as the characteristics of city and region, infrastructure and technical parameters, existing connections, institutional circumstances, and capacity, are evaluated on a sliding scale. This is a rough analysis used by planners while evaluating whether a city is suitable for a chosen rail transportation system [19], but it can be suitable for local-level projects where qualitative effects are highly relevant. One of the key advantages of an MCA is that it can easily include stakeholder participation, while experts, decision-makers, and public institutions can be involved in the performance scoring and weighting of criteria.

Recent research on the perceived strengths and weaknesses of CBA and MCA, and the ability of each method to support sustainable transportation decision processes, showed that, by adopting a more global and holistic perspective and by facilitating the inclusion of a participative process, MCA (or a combination of CBA and MCA), is a more promising appraisal method for sustainable transportation [20]. On the other hand, this means that MCA can be very subjective, i.e., the outcomes can be biased [21].

Although the discourse of sustainable mobility has led to new policies for managing urban and suburban mobility through the development of new and/or the rehabilitation of existing rail infrastructure, the literature review showed that there is no universal, directly applicable methodology for evaluating such projects, due to numerous site-specific conditions. To provide guidance for future urban rail infrastructure feasibility studies in similar conditions, this paper aims to contribute to the process of the evaluation of urban rail infrastructure projects through the presentation of the methodology and the results of a preliminary feasibility study (PFS) of the revitalization, development, and (re)integration of the rail with road, maritime, and air transportation in the ZUA. This contribution will be reflected in the following: (1) the proposal of sustainable urban rail project evaluation criteria (concerning environmental, economic, engineering, social, and safety aspects), (2) the proposal of evaluation criteria indicators, and (3) the proposal of an indicators evaluation criteria quantification process (concerning data availability and level of detail).

The investigation presented in this paper was performed in four steps. The first step was to define the vision of the future transportation system in the ZUA and then to define the goals to be achieved and measures to be performed to make it come to life. The second step was to design multiple variants of rail infrastructure required for achieving the identified vision. The third step was to evaluate the designed variants by multi-criteria analysis supplemented with partial cost–benefit analysis, prioritize them, and select the most preferable one, i.e., the one to be further pursued and elaborated in the ZUA's transportation system development plans. Additionally, to reduce the influence of stakeholders' bias, i.e., to ensure a high level of objectivity, auditability, and transparency in the evaluation process, and to assess how the PFS criteria prioritization influenced the results of a proposed decision-making model, a sensitivity analysis was performed where the criteria weights were systematically varied to observe how sensitive or responsive the PFS results are to such changes.

## 2. Materials and Methods

In this section, the proposed evaluation model is described in detail, while the evaluation methodology is presented in Figure 1.

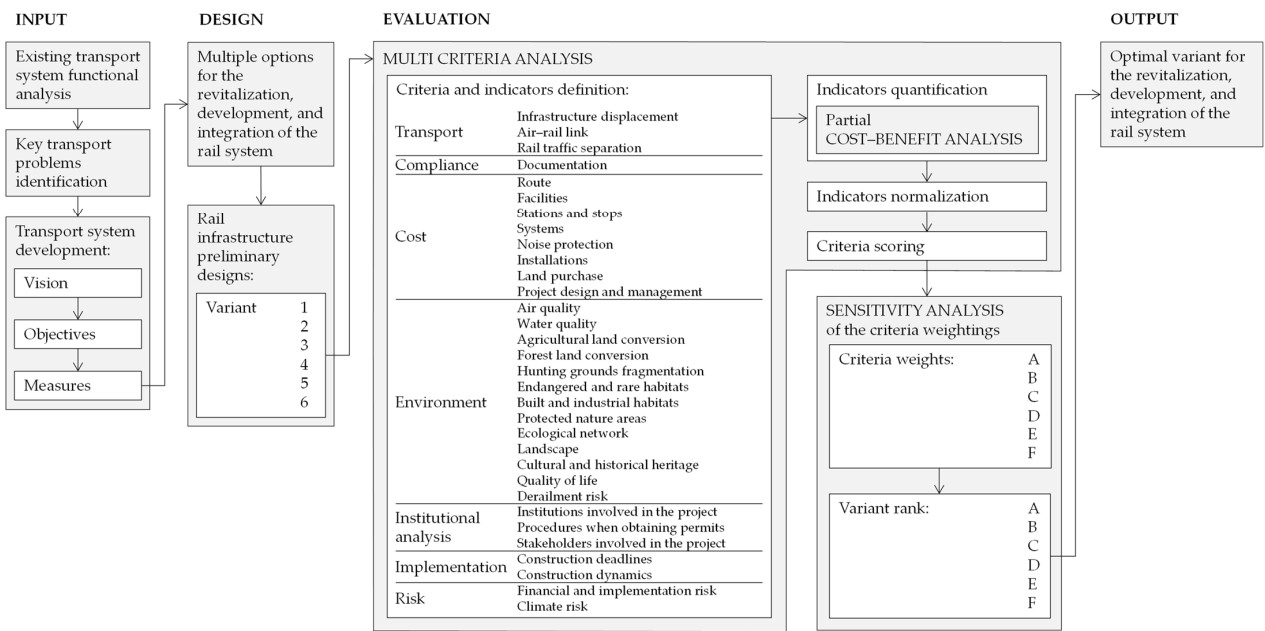

**Figure 1.** Investigation methodology.

To define a vision of the ZUA's transportation system development, a functional analysis of the existing transportation system was performed. This analysis of existing traffic demand was based on a set of data from secondary, publicly available sources and documents.

According to [22], the ZUA, spreading over 790 km$^2$, consists of two cities (Zadar and Nin) and 13 municipalities. According to the population census conducted in 2021 [23], the average population density is almost twice the national average, and 65% of 108,198 inhabitants live in the city of Zadar where most of the activities and services (social, administrative, health, educational, and recreational institutions) are concentrated. Considering the share of daily migrations to Zadar presented in [24], the ZUA can be divided into four zones: (1) a continuous coastal urban and urbanized belt, which includes the municipalities that are functionally closely connected to Zadar, with more than 50% of daily migrants; (2) the first belt of landlocked municipalities with more than 50% of daily migrants; (3) the second belt of landlocked municipalities with 30 to 50% of daily migrants; and (4) municipalities on the islands with less than 30% of daily migrants.

According to [25], the ZUA's economy is dominated by trade, processing industry, maritime transportation, mariculture, and tourism. It accounts for approximately 3% of the national economy. The share in the national branch economy is very high for three specific sectors: maritime transportation (up to 40%), mariculture (over 70%), and the tourism sector (up to 14%). According to [22,24], the greatest potential for the ZUA's further economic growth lies in tourism and complementary activities. The ZUA tourist sector development has been stimulated by the improvement of communal, tourist, and transportation infrastructure shown in Figure 2, especially the connection with European transportation corridors via the A1 highway in 2005, the expansion of the passenger terminal building of Zadar Airport, together with the introduction of low-cost airlines connecting Zadar with Central and Western Europe in 2007, and the construction of Gaženica Port for domestic and international ferry transportation and cruisers in 2015.

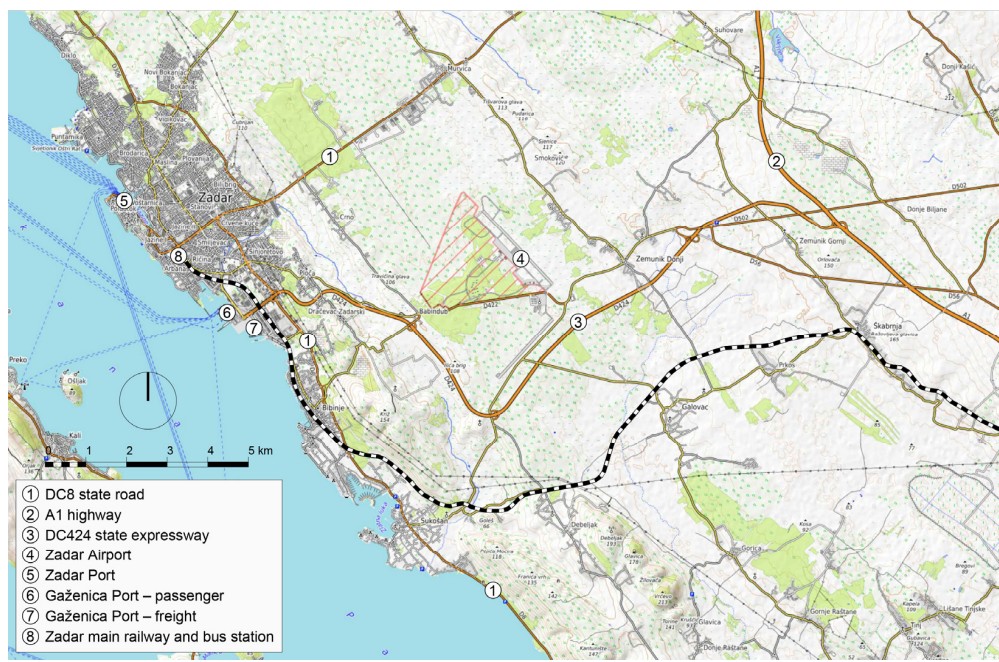

**Figure 2.** ZUA transportation infrastructure [2].

The backbone of the road transportation system in the ZUA is the state road DC8. It connects the main settlements and Zadar with the A1 highway and other important settlements on the Croatian coast. Also, the four-lane expressway DC424 connects Zadar, Gaženica Port (passenger and freight), Zadar Airport, and the A1 highway.

According to [26], 86 public buses maintain a schedule of a total of 37 lines with 485 departures per day. They transport around 8 million passengers annually, of which 75% are transported on urban, 20% on suburban, and 5% on island bus lines. The most frequent bus line passes through the central part of Zadar, and one line connects Zadar with Zadar Airport.

Zadar Airport is located 10 km east of Zadar. According to [27], passenger air traffic is highly seasonal and constantly increasing, with 1.1 million passengers in 2022 (30% more than the record from 2019, Figure 3). Traffic between the ZUA and Zadar Airport takes place on only one access road, and only 10% of the passengers use public bus transportation. Freight air traffic is occasional and almost negligible in quantities, as shown in Figure 4.

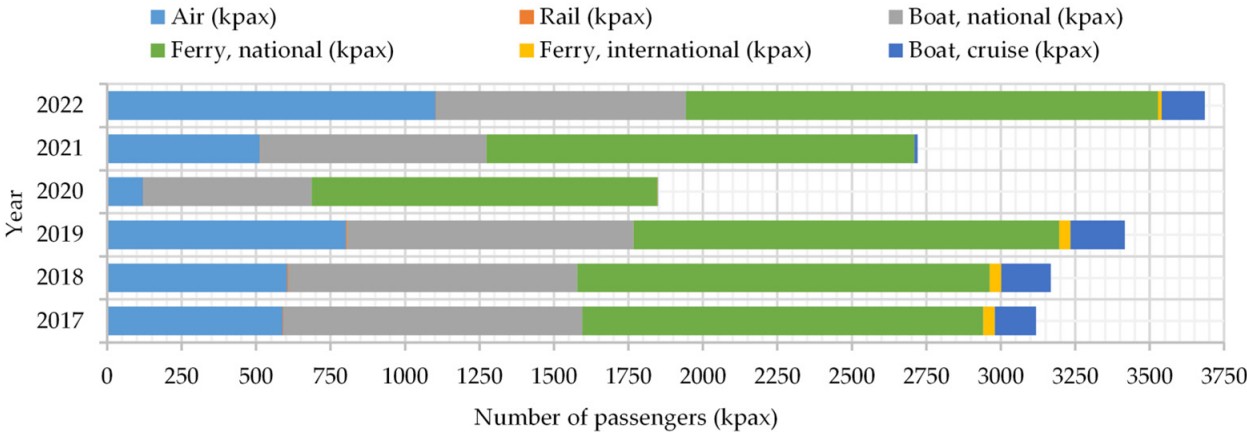

**Figure 3.** Annual volume of passengers in thousands of passengers based on transportation mode, plotted by the authors according to the data given in [23,27,28].

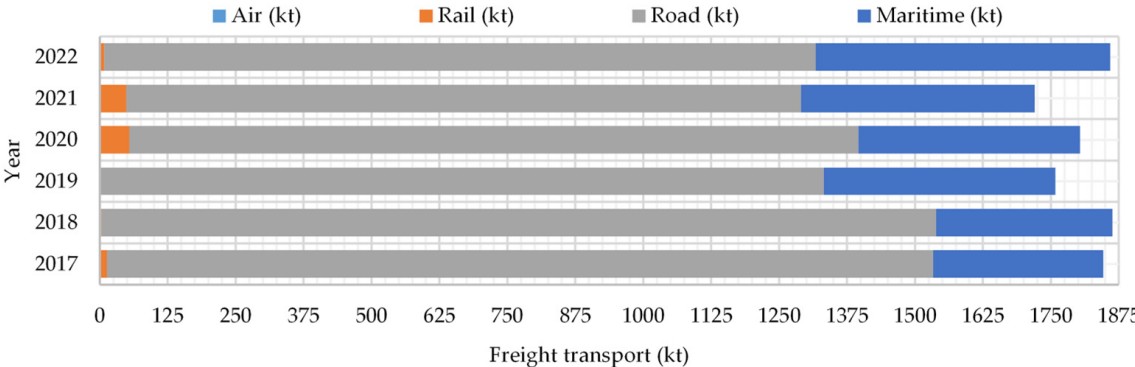

**Figure 4.** Annual cargo volume in thousands of tons based on transportation mode, plotted by the authors according to the data given in [23,27,28].

The maritime transportation system in the ZUA consists of two ports. Zadar Port for national passenger ship transportation is situated in the Zadar City center (peninsula). It connects Zadar with the islands of the Zadar archipelago and Kvarner with nine national shipping lines. It is connected to DC8 through a dense urban fabric. Gaženica Port consists of a passenger, freight, and fishing port. The ferry port connects Zadar with the islands of the Zadar archipelago and Kvarner with five state ferry lines. Its launch in 2015 relieved the passenger port in the Zadar peninsula. The international ferry line Zadar–Ancona and cruisers operate there in the summer months. Today, the number of transported passengers and vehicles in international traffic is stagnating, while the number of passengers from ships on cruises is increasing significantly from year to year, as shown in Figure 3. The cargo port of Gaženica enables the transshipment of liquid, bulk, and general cargo, and although there is no possibility of accepting container cargo, this is the only port on the Adriatic that can accept and deliver special cargo. It is the main generator of road freight transportation in the area, as shown in Figure 4 [28].

The Zadar railway station for international, national, and local rail transportation is situated in the center of the city next to the bus terminal. The rail line constructed in the 1960s continues via Knin through two of the ZUA's most developed municipalities (Bibinje and Sukošan) and Gaženica Port. Due to years of neglect, i.e., lack of infrastructure development, modernization, and maintenance, it is in an unsatisfactory technical condition. In 2014, it was declared unprofitable for passenger transportation and closed for traffic. Today, passenger transportation between railway stations in the ZUA is carried out by buses, and freight traffic is negligible, as shown in Figures 3 and 4 [23].

According to [29], PC is the dominant mean of transportation in the area, while public transportation is used to a greater extent only by students, as shown in Figure 5. Despite the investments in tourist, communal, and transportation infrastructure, tourism in the ZUA is still extremely seasonal. Namely, over half of all tourist arrivals occur in just two summer months, with over 150,000 arrivals in the summer of 2022 [30]. The large number of tourists in July and August significantly affects traffic demand and the traffic network load, as shown in Figure 6 [29].

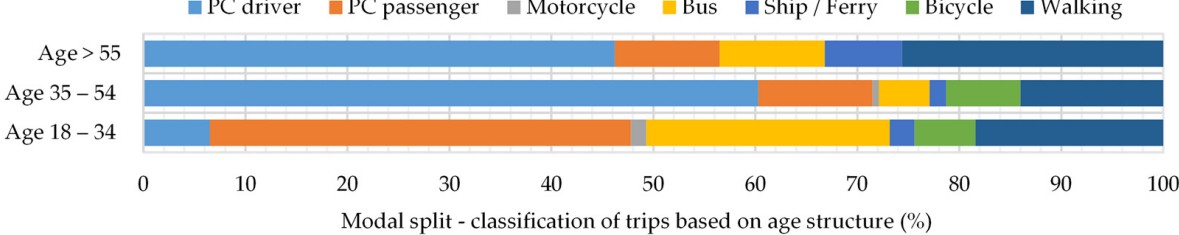

**Figure 5.** Modal split based on population age structure, plotted by the authors according to the data given in [29].

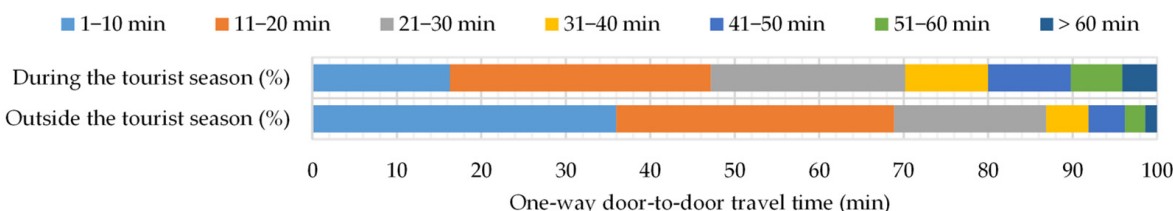

**Figure 6.** One-way door-to-door travel time in minutes outside of and during the tourist season, plotted by the authors according to the data given in [29].

Determining existing traffic demand and its impact on the ZUA's traffic supply identified the area's key transportation problems. After defining the PFS vision for solving these problems, an analysis of the priorities and the objectives given in strategic documents at the European [31–33], national [34–36], regional [29,37], and local levels [22,24,38,39] was carried out to define PFS objectives compliant with this legal framework. To identify rational measures for ensuring the defined objectives, spatial planning documents at regional, city, municipal, and local scales, masterplans for maritime freight and passenger air transportation infrastructure development, and available preliminary designs of railway infrastructure upgrades were analyzed.

The second step was to define multiple options for revitalization, development, and integration of the rail system with road, maritime, and air transportation in the ZUA. This included the preliminary design of six variants of the railway routes. The existing railway line is a conventional single-track railway line for mixed traffic and a part of the comprehensive EU TEN-T network. As it is potentially important for international traffic, its redesign followed interoperability conditions. The design had to satisfy the following three conditions:

- New railway links were intended for local passenger traffic and mixed traffic.
- Due to spatial constraints, the location of the rail cargo terminal for Gaženica Port defined in the preliminary design of the bypass of the settlement of Bibinje and the freight station Gaženica [40] was non-negotiable.
- The line from Gaženica Port to Zadar Station was to be for passenger traffic only, and, due to the high degree of urbanization, potential interventions in its route had to be minimal.

The six variants were designed in four realization phases, according to their basic, technological purpose. Phase I diverted the rail line from coastal municipalities, and was designed as a single-track railway for international mixed traffic, with speeds up to 120 km/h and grades up to 12.5 mm/m. Phase II enabled the separation of freight and international passenger transportation and was designed as a single-track railway for international passenger traffic, with speeds up to 80 km/h and grades up to 12.5 mm/m. Phase III enabled the link between Gaženica Port and a logistics center with storage and transshipment capacities in the vicinity of Zadar Airport, and was designed as a single-track railway for local mixed traffic, with speeds up to 100 km/h and grades up to 20 mm/m. Phase IV enabled the link between Gaženica Port and Zadar Airport, and was designed as a single-track railway for suburban and urban passenger traffic, with speeds up to 50 km/h and grades up to 35 mm/m.

The third step was to evaluate the designed six rail route variants. During the process of evaluation method selection, it was observed that variants' design approached the ZUA's transportation issues from different angles. This meant that variants could provide different benefits, and contribute differently to the area's economic, social, and territorial issues. Also, already planned interventions in transportation systems and infrastructure, used as input for variant designs, were at different development stages, ranging from the ones that were only at the conceptual design stage to the partially built infrastructure. Therefore, it was decided that if the variants were to be evaluated by a standard CBA approach, the results could be misleading, i.e., they could lead to ineffective long-term infrastructure

decisions. This specific situation called for the consideration of multidisciplinary and multidimensional aspects of the prioritization of the ZUA's railway system revitalization, development, and integration options. For that purpose, the MCA approach seemed most suitable.

According to [41], there are two basic approaches to MCA and CBA in the process of transportation infrastructure evaluation. The first involves conducting MCA instead of CBA. It is usually implemented if there are no sufficient or reliable data for the reasonable completion of CBA. Such use of MCA takes place when criteria other than economic and financial are critical, significantly impact project assessment, and reflect beneficiaries' or investors' preferences. The second approach treats CBA as a component of MCA. It involves conducting CBA to the level that is allowed by data availability, and the utilization of CBA results as inputs and elements of an MCA. Today, such an approach to MCA is frequently used for prioritizing public transportation infrastructure development variants, especially those that could be funded by international donors, as is the case in the ZUA.

The performance of MCA, i.e., the process of assigning scores to the developed variants of (re)integrating rail with road, maritime, and air transportation in the ZUA, involved defining the variants' evaluation criteria and their indicators, defining the indicators' values, their normalization, score calculation for each criterion, assigning weights to the criteria, and calculation of final, aggregated scores for each evaluated variant. Seven evaluation criteria (namely: transport, compliance, costs, environment, institutional analysis, implementation, and risk, each described by several indicators), were defined based on the literature review [42–44] and previous experiences of the authors.

The "Transport" criterion was used to evaluate the impact of a particular variant on the existing transportation system based on the following indicators, expressed in track length:

- Displacement of railway infrastructure from the coastal municipalities of Bibinje and Sukošan, i.e., from the areas with high potential for further urban and economic development,
- Creation of the rail link between the Zadar railway station and Zadar Airport,
- Separation of slow and heavy freight from fast and light passenger rail transportation.

The "Compliance" criterion was used to evaluate the compliance of a particular variant with current strategic and spatial planning documents. The criterion indicator was defined as the number of current documents the variant complies with.

The "Costs" criterion was used to evaluate the investment costs of a particular variant calculated based on [45]. The indicators, i.e., direct investment costs of a particular variant including construction, land acquisition, design, supervision, and consulting, were defined as the eight following basic investment costs, expressed in currency:

- Route (track superstructure and substructure) construction,
- Facilities (bridges, viaducts, tunnels, underpasses, overpasses) construction,
- Stations and stops (track structure, buildings, and platforms) construction,
- Systems (electrification, signaling and safety, telecommunications, and central traffic management) construction and implementation,
- Noise barrier construction,
- Communal installations construction,
- Land purchase,
- Project design and management.

The "Environment" criterion was used to analyze and evaluate the impact of a particular variant on the environment within the area extending 250 m from the railway line, and environmental protection measures that should be applied given the identified impacts. The twelve indicators of the variants' impact on the environment, expressed in track length, were the following:

- Air quality, expressed as a moderate increase in the concentration of pollutants in the air along the new routes, outside tunnels, and in the vicinity of tunnel portals,

- Pollution of surface, underground, and coastal water bodies, and sanitary protection zones due to the possible discharge of polluted water from trains, and in case of accidental situations or improper technical maintenance of the system, along the length of the new routes,
- The required length of conversion of agricultural areas to infrastructural areas,
- Required conversion of forests to infrastructural areas,
- Length of fragmentation of hunting grounds,
- Length of variant passing through endangered and rare habitats, and industrial habitats, thus affecting biodiversity,
- Length of variant passing through protected nature areas,
- Length of ecological network impacted by variant,
- Landscape anthropogenicity along the variant route,
- Cultural and historical heritage locations along the variant route in phase II,
- Number of people with reduced quality of life along the variant route in phase II,
- Risk of derailment of freight trains transporting substances harmful and dangerous to the environment along the variant route.

The "Institutional analysis" criterion was used to evaluate the impact of a particular variant regarding the following three indicators, expressed as a number:

- Various institutions necessary to be involved in the project,
- Necessary procedures when obtaining permits,
- Different stakeholders required for the realization of a particular variant.

The "Implementation" criterion was used to evaluate the impact of a particular variant on project construction deadlines and construction dynamics. The measure that was chosen for the construction deadlines indicator was the length of the track, and, for the construction dynamics, the proportion of facilities along the variant route.

The "Risk" criterion was used to evaluate the sensitivity of a particular variant to financial and implementation risks, and climate risks. The measure that was chosen for the financial and implementation risks indicator was the average deviation, i.e., mean absolute deviation, of variant phase length. Generally, the average deviation is calculated similarly to the standard deviation, but it uses absolute values instead of squares to circumvent the issue of negative differences between the data points and their means. To calculate the average deviations of phases' route lengths, the mean length of all four route phases was calculated for each variant. Then, the difference between the mean length and length of each phase was calculated. Finally, the average of the absolute values of those differences was determined. The measure that was chosen for the climate risk indicator was the proportion of tunnels along variant routes, calculated by dividing the length of the track route in tunnels by the total variant length.

To simplify the procedure, indicators of a criterium that were expressed in identical measure units (costs, environment, and institutional analysis criterion indicators) were summed up for each variant, and the single, total indicator value was defined.

After defining the indicator values for all six variants, scores from 1 to 5 were chosen for their normalization. Table 1 shows the ranges of indicator values and associated scores. Score 5 was considered the highest score, and it was achievable for the design that could enable the shortest length of existing rail line displacement from two coastal municipalities, the shortest link between Zadar railway station and Zadar Airport, the total separation of passenger rail transportation from Zadar Station to Zadar Airport, full availability of planning documents, the smallest total investment costs, no impact on the environment, the involvement of the smallest number of procedures, institutions, and stakeholders, the shortest construction deadlines, i.e., shortest total track length, the easiest construction dynamics, i.e., no facilities (tunnels, bridges, etc.) along the route, the smallest financial and implementation risks, i.e., the most uniform length of track route phases, and the smallest climate risk, i.e., the placement of most of the tracks underground.

**Table 1.** Ranges of indicator values and associated scores used in the variants' evaluation.

| Criterium | Indicator | Measure Unit | Score | | | | |
|---|---|---|---|---|---|---|---|
| | | | 1 | 2 | 3 | 4 | 5 |
| Transport | Track displacement length | (km) | 14.25–17.50 | 11.00–14.25 | 7.75–11.00 | 4.50–7.75 | ≤4.50 |
| | Zadar–Airport link length | (km) | 17.5–20.5 | 14.5–17.5 | 11.5–14.5 | 8.5–11.5 | ≤8.5 |
| | Traffic separation on Zadar–Airport link | (%) | 0–25 | 25–50 | 50–75 | 75–100 | 100 |
| Compliance | Strategic, planning and project documents | (no) | 0–3 | 3–6 | 6–9 | 9–12 | ≥12 |
| Cost | Total investment costs | (mil. EUR) | 200–225 | 175–200 | 150–175 | 125–150 | ≤125 |
| Environment | Track affecting the environment | (km) | 200–250 | 150–200 | 100–150 | 50–100 | ≤50 |
| Institutional analysis | Institutions, procedures, stakeholders | (no) | 9–11 | 7–9 | 5–7 | 3–5 | ≤3 |
| Implementation | Construction deadlines | (km) | 30–35 | 25–30 | 20–25 | 15–20 | ≤15 |
| | Construction dynamics | (%) | 75–100 | 50–75 | 25–50 | 0–25 | ≤0 |
| Risk | Financial and implementation risks | (km) | 3.0–3.5 | 2.5–3.0 | 2.0–2.5 | 1.5–2.0 | ≤1.5 |
| | Climate risk | (%) | 0–20 | 20–40 | 40–60 | 60–80 | ≥80 |

The normalized indicator scores for each variant were determined by linear interpolation. The scores of indicators that were expressed in different measurement units (transport, implementation, and risk criterion indicators) were averaged to obtain the criterium score. The criteria weight ratios presented in Table 2 as weight combination A were determined through a consultation process in which key stakeholders took part. The individual evaluation criteria scores were weighted and the result of the evaluation of each variant was defined as the sum of the weighted evaluation criteria scores.

**Table 2.** Criteria weight combinations that were determined through a consultation process (A) and used in the sensitivity analysis (B to F).

| Criteria | Criterium Weight (%) Combinations | | | | | |
|---|---|---|---|---|---|---|
| | A | B | C | D | E | F |
| Transport | 30 | 12 | 12 | 12 | 12 | 12 |
| Compliance | 12 | 30 | 12 | 12 | 12 | 12 |
| Cost | 12 | 12 | 30 | 12 | 12 | 12 |
| Environment | 12 | 12 | 12 | 30 | 12 | 12 |
| Institutional analysis | 12 | 12 | 12 | 12 | 30 | 12 |
| Implementation | 12 | 12 | 12 | 12 | 12 | 30 |
| Risk | 10 | 10 | 10 | 10 | 10 | 10 |

In the fourth step, the sensitivity analysis was performed to determine the robustness of the decision-making process. In general, sensitivity analysis is a method for analyzing the effect of uncertainty in the output of a system, subject to uncertainties in the inputs. To observe the effect of uncertainties, the inputs to the system are varied and their corresponding effect on the outputs is studied [46]. In this case, the inputs were the weightings of the criteria, and the output was the ranking of development variants. This allowed the influence of changes in the weights of criteria (presented in Table 2, as weight combinations B to F) on the variants' evaluation scores to be tested. To simplify the procedure, the 30% weight was varied systematically between transportation, compliance, cost, environment, institutional analysis, and implementation criteria, and the risk criterion in all analyzed weight combinations remained unchanged, at 10%.

## 3. Results

### 3.1. Vision, Objectives, and Measures

The analysis of the socio-economic context and the functional analysis of the existing ZUA's transportation system identified the following issues: Due to the concentration of activities and services in Zadar, the dense and high-quality road network, and the high degree of motorization, PC is the first choice for daily migrations. The mobility of the local population and freight transportation is limited during the summer months when travel

time for short, under 10 km, road trips in the ZUA doubles due to the tourists. The main problem is the lack of alternative connections between the Zadar city center, sea passenger port, and airport.

Considering the trend of investments in the tourism sector, the planned development of the Zadar Airport, Gaženica Port, and the industrial zones in their vicinity, it can be expected that the problem of overloading the road network will spill over into the rest of the year. Regardless of the parameters resulting from the favorable geographical-traffic position of the ZUA, the existing transportation system does not correspond to the modern demands and needs of the population and the economy. Even though the potential of intermodal transportation is extremely high, the transportation system is not integrated, and could easily become a limiting factor in the overall development of the ZUA. Based on this conclusion, the following vision for the further development of the transportation system of the ZUA was formed: *"Increasing accessibility and mobility within the area can be achieved by increasing the quality of the transport network through the integration of four key components: bus, sea, air, and rail transport. The organization of the intermodal transport system would enable the introduction of uniform tariffs, better organization of traffic, decongestion of road infrastructure, energy savings, environmental protection, and better functional integration of the area. It would reduce the burden on current parking capacities, mostly in Zadar, and the City would finally be able to organize pedestrian and bicycle zones. This would reduce traffic congestion, increase safety, and free up space for environmentally friendly zero-emission mobility solutions".*

The results of the performed compliance analysis of the PFS objectives with the legal framework are given in Appendix A, Tables A1–A4. The analysis of European, national, regional, and local legal frameworks resulted in defining the following seven objectives of the PFS of revitalization, development, and integration of the rail system in the ZUA:

- Revitalization of the existing railway infrastructure,
- Relocation of freight rail traffic from the narrow and densely developed suburban coastal area,
- Promotion of intermodal passenger and freight transportation,
- Improvement of urban and regional accessibility and connectivity,
- Increase of traffic safety,
- Reduction of travel time and operating costs,
- Reduction of traffic impacts on the environment.

The analysis of spatial planning documents on regional, city, municipal, and local scales, masterplans for maritime freight and passenger air transportation infrastructure development, and preliminary designs of railway infrastructure upgrades defined the conditions, guidelines, and measures for land allocations (Figure 7) influencing the future distribution of people and activities in the ZUA by planning the (1) reconstruction of the existing and construction of new maneuvering areas of Zadar Airport to increase the number of international flights, (2) construction of a logistics center with storage and transshipment capacities in the vicinity of Zadar Airport, (3) construction of road infrastructure to connect air and sea passenger and cargo traffic, (4) construction of an industrial and storage area next to Gaženica Port, (5) construction of terminal infrastructure in Gaženica Port for intermodal (ship, rail, and road) container transportation, (6) construction of the Gaženica freight railway station next to Gaženica Port, and (7) modernization of the existing railway line and its dislocation from Bibinje.

The simultaneous analysis of the abovementioned documents showed that they did not recognize the full potential of the existing railway system, despite the planned construction of new capacities for rail freight transportation and existing railway reconstruction and dislocation from the high-value coastal area. It was concluded that, to realize the objectives for improving mobility in the ZUA, the construction of a rail link between the Zadar railway station, Zadar Airport, and Gaženica Port should be investigated, as it could fully realize the intermodal connection of the area. This link, in addition to being able to take over freight traffic from the roads, could also restore and improve the passenger function that

residents and tourists need for daily migrations and ensure the sustainability of the area's transportation system.

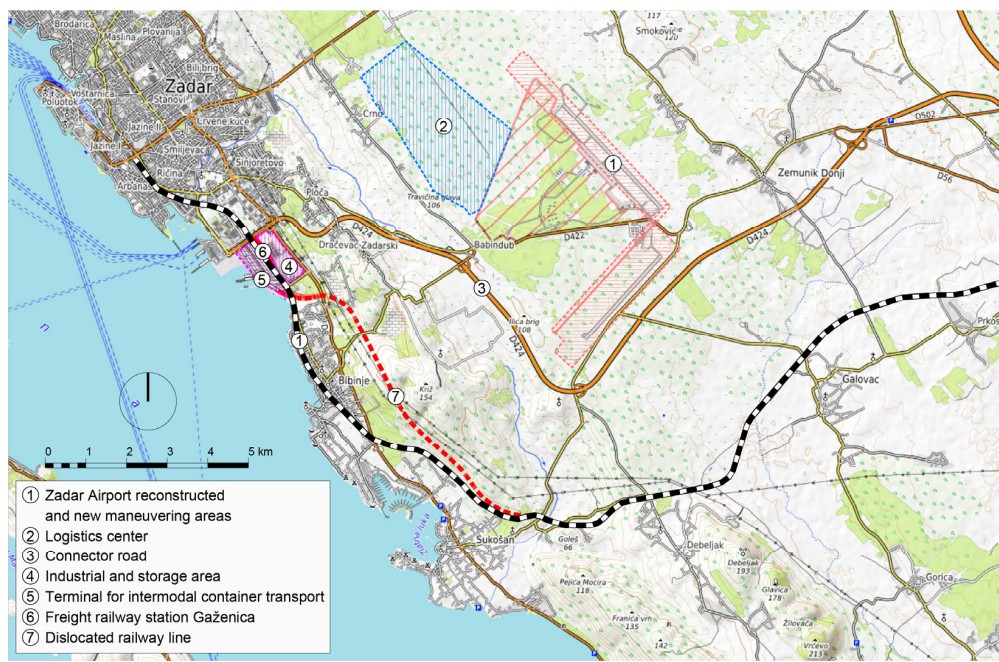

**Figure 7.** Planned infrastructure development in ZUA [2].

## 3.2. Variant Development

In all six designed rail route alignment variants shown in Figure 8, the design of the planned Gaženica freight station was adopted from the preliminary design of the bypass of the settlement of Bibinje and the Gaženica freight station [40]. The line from the Gaženica junction to the Zadar railway station and the Gaženica Port stop was designed for passenger traffic only.

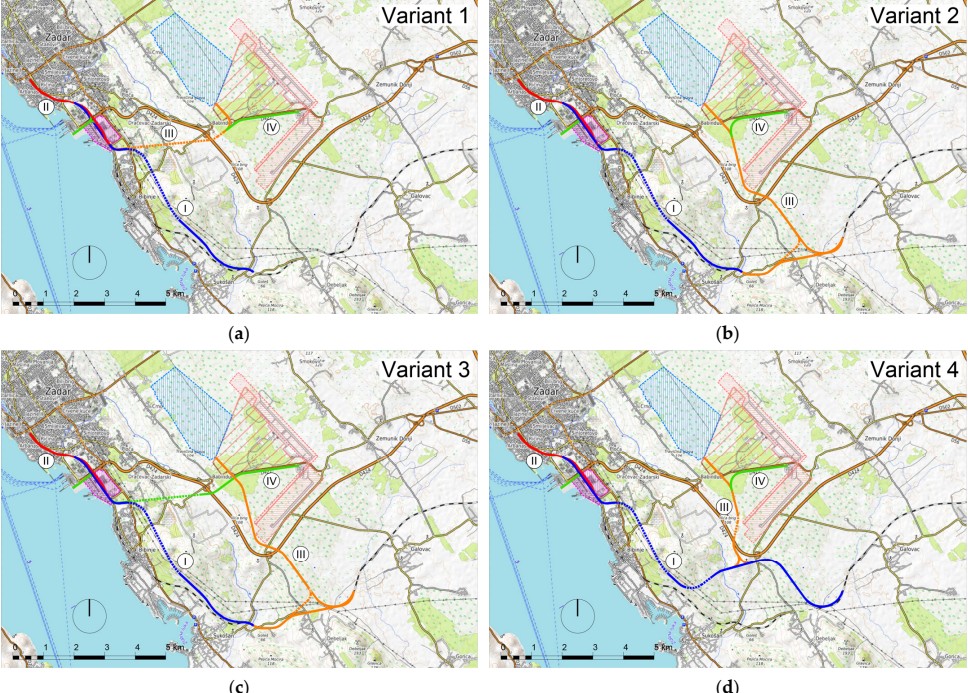

**Figure 8.** *Cont.*

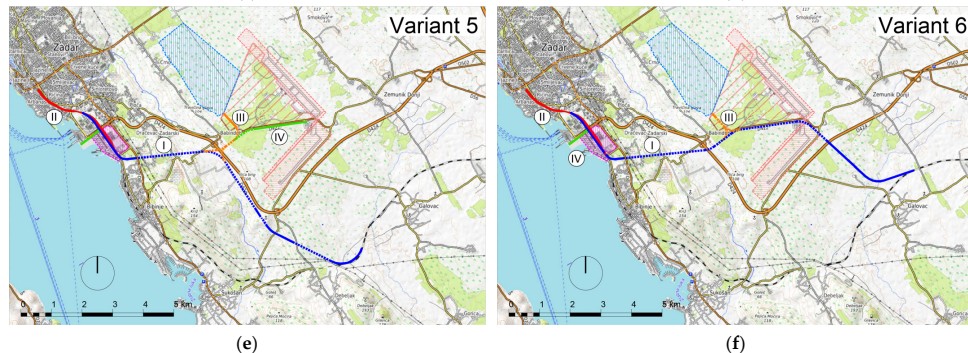

(**e**)　　　　　　　　　　　　　　　　(**f**)

**Figure 8.** Horizontal alignment of six proposed variants of railway routes in ZUA, Phase I shown in blue, Phase II shown in red, Phase III shown in orange, and Phase IV shown in green: (**a**) Variant 1; (**b**) Variant 2; (**c**) Variant 3; (**d**) Variant 4; (**e**) Variant 5; (**f**) Variant 6 [2].

In Variant 1, Variant 2, and Variant 3, the rail route was diverted from Bibinje and Sukošan, according to [40]. In Variant 1, the rail line to the logistics center and Zadar Airport was designed for mixed traffic until the Babindub junction. In Variant 2, the connection to the logistics center and Zadar Airport was achieved by a single-track for mixed traffic, until the Babindub junction, branching off from the existing rail line in Galovac. In Variant 3, the connection to the logistics center and Zadar Airport was designed as a combination of the designs given in Variant 1 and Variant 2.

In Variant 4 and Variant 5, the rail route was diverted from Bibinje, Sukošan, and Galovac. In Variant 4, the rail connection from Sveti Martin to the logistics center and Zadar Airport was shorter than the one in Variant 3, due to the extensive intervention in the existing railway route. In Variant 5, the connection to the logistics center and Zadar Airport is shorter than the one in Variant 4, due to the even more extensive intervention in the existing railway route.

In Variant 6, the rail route was diverted from Bibinje, Sukošan, Galovac, and Škabrnja. The connection to the logistics center was designed as the most extensive intervention in the existing rail route. The Zadar Airport stop was designed as a transit stop on this bypass route.

### 3.3. Multi-Criteria Analysis

The results of the indicator values calculation are presented in Table 3. Figures 9–11. show calculated values of costs, environment, and institutional analysis criteria indicators that were expressed in identical measure units and summed up for each variant to define a total indicator value given in Table 3. Normalized indicator values are presented in Table 4, and calculated criterium scores are presented in Table 5.

**Table 3.** Calculated variants indicator values.

| Criterium | Indicator | Measure Unit | Variant | | | | | |
|---|---|---|---|---|---|---|---|---|
| | | | 1 | 2 | 3 | 4 | 5 | 6 |
| Transport | Track displacement length | (km) | 6.7104 | 6.7104 | 6.7104 | 10.6149 | 9.815 | 11.055 |
| | Zadar–Airport link length | (km) | 9.980 | 19.153 | 9.9825 | 14.636 | 9.980 | 9.981 |
| | Traffic separation on Zadar–Airport link | (%) | 25 | 12 | 65 | 16 | 25 | 0 |
| Compliance | Strategic, planning and project documents | (no) | 8 | 8 | 8 | 6 | 4 | 4 |
| Cost | Total investment costs | (mil. EUR) | 138.750 | 186.465 | 194.447 | 197.431 | 143.332 | 196.406 |
| Environment | Track affecting the environment | (km) | 122.327 | 186.103 | 201.072 | 159.496 | 114.074 | 96.475 |
| Institutional analysis | Institutions, procedures, stakeholders | (no) | 6 | 6 | 6 | 8 | 8 | 10 |
| Implementation | Construction deadlines | (km) | 21.771 | 28.993 | 33.166 | 24.511 | 20.753 | 18.442 |
| | Construction dynamics | (%) | 19 | 14 | 19 | 21 | 49 | 47 |
| Risk | Financial and implementation risks | (km) | 1.5 | 3.2 | 2.2 | 3.1 | 2.3 | 3.2 |
| | Climate risk | (%) | 29 | 13 | 19 | 20 | 43 | 43 |

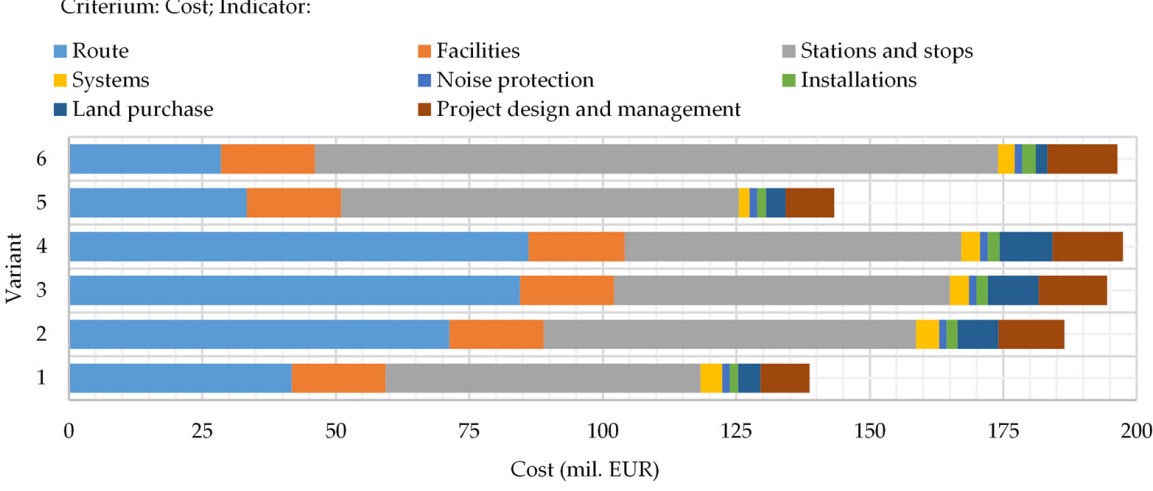

**Figure 9.** Cost criterium indicator values.

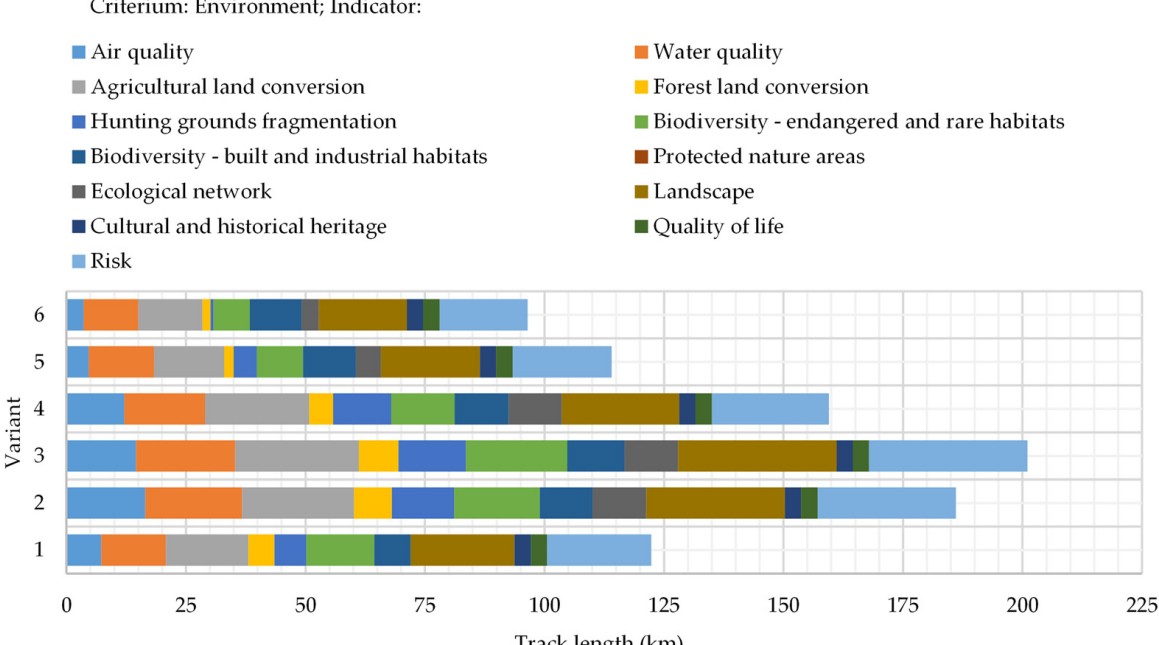

**Figure 10.** Environment criterium indicator values.

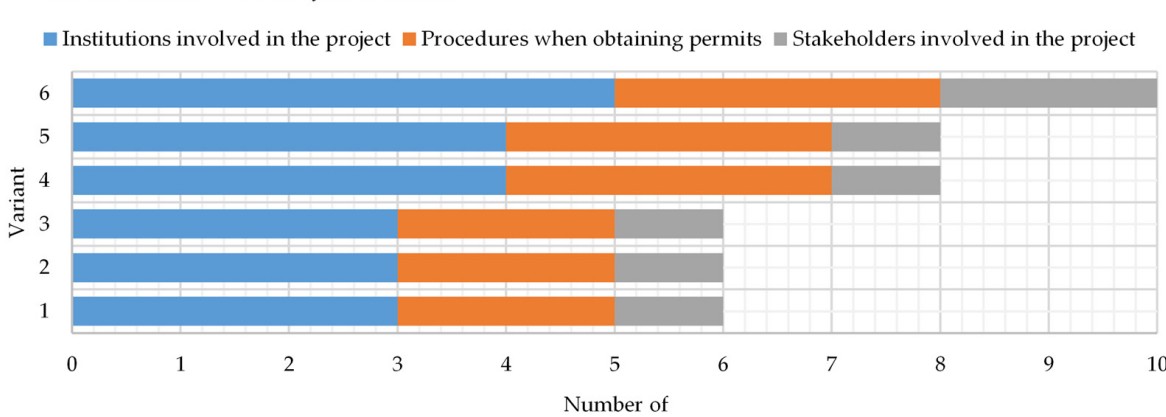

**Figure 11.** Institutional analysis criterium indicator values.

**Table 4.** Normalized indicator values.

| Criterium | Indicator | Variant | | | | | |
|---|---|---|---|---|---|---|---|
| | | 1 | 2 | 3 | 4 | 5 | 6 |
| Transport | Track displacement length | 4.3 | 4.3 | 4.3 | 3.1 | 3.4 | 3.0 |
| | Zadar–Airport link length | 4.5 | 1.5 | 4.5 | 3.0 | 4.5 | 4.5 |
| | Traffic separation on Zadar–Airport link | 2.0 | 1.5 | 3.6 | 1.6 | 2.0 | 1.0 |
| Compliance | Strategic, planning and project documents | 3.7 | 3.7 | 3.7 | 3.0 | 2.3 | 2.3 |
| Cost | Total investment costs | 4.5 | 2.6 | 2.2 | 2.1 | 4.3 | 2.1 |
| Environment | Track affecting the environment | 3.6 | 2.3 | 2.0 | 2.8 | 3.7 | 4.1 |
| Institutional analysis | Institutions, procedures, stakeholders | 3.5 | 3.5 | 3.5 | 2.5 | 2.5 | 1.5 |
| Implementation | Construction deadlines | 3.7 | 2.2 | 1.4 | 2.1 | 3.9 | 4.3 |
| | Construction dynamics | 3.8 | 4.4 | 4.2 | 4.2 | 3.0 | 3.1 |
| Risk | Financial and implementation risks | 4.9 | 1.5 | 3.6 | 1.7 | 3.4 | 1.6 |
| | Climate risk | 2.5 | 1.7 | 2.0 | 2.0 | 2.2 | 2.2 |

**Table 5.** Criterium scores.

| Criteria | Variant | | | | | |
|---|---|---|---|---|---|---|
| | 1 | 2 | 3 | 4 | 5 | 6 |
| Transport | 3.6 | 2.4 | 4.1 | 2.6 | 3.3 | 2.8 |
| Compliance | 3.7 | 3.7 | 3.7 | 3.0 | 2.3 | 2.3 |
| Cost | 4.5 | 2.6 | 2.2 | 2.1 | 4.3 | 2.1 |
| Environment | 3.6 | 2.3 | 2.0 | 2.8 | 3.7 | 4.1 |
| Institutional analysis | 3.5 | 3.5 | 3.5 | 2.5 | 2.5 | 1.5 |
| Implementation | 3.7 | 3.3 | 2.8 | 3.1 | 3.4 | 3.7 |
| Risk | 3.7 | 1.6 | 2.8 | 1.9 | 2.8 | 1.9 |

As Table 5 shows, the results of the analysis according to the transport criterion showed that Variant 3 and Variant 1 enabled the greatest traffic effects, considering that the construction of the rail bypass diverted freight traffic from the coastal settlements and created the shortest rail link to Zadar Airport. Variant 2 and Variant 5 scored lower because they did not provide a direct link to Zadar Airport, which resulted in longer travel times. Variant 6 was also scored lower, as passenger and freight traffic largely shared the traffic corridor. According to the compliance criterion, Variant 1, Variant 2, and Variant 3 enabled the greatest compliance, especially in the initial stages, considering that the planned rail bypass was compliant with strategic documents, incorporated the spatial planning documentation, and had a completed conceptual project. Variant 4 was partially compliant with a completed rail bypass conceptual project and scored lower, while Variant 5 and Variant 6, apart from compliance with strategic documents, were not compliant with spatial planning documentation or the bypass conceptual project and were scored the lowest. According to the cost criterion, Variant 1 and Variant 5 enabled the smallest total investment costs. As for Variant 6, the costs for stations and stop construction were extremely high considering that the planned intervention proposed an underground Zadar Airport stop. According to the environment criterion, variants with a significant percentage of tunnels received the highest scores due to the reduced impact on the environment. Considering that all variants included construction in new, unbuilt corridors, the scores were relatively low. According to the institutional analysis criterion, Variant 1, Variant 2, and Variant 3 received higher scores. As the implementation phasing was foreseen for all six variants, the number of required building permits increased. Variant 6 received the lowest score, considering that the inclusion of the Zadar Airport authority was necessary, as the variant passed through the airport area. According to the implementation criterion, Variant 1 received the highest average score, given that its combined track and facility lengths

were the shortest, which significantly affected the construction deadlines and dynamics. Finally, the results of the analysis according to the risk criterion showed that the variants with a significant percentage of tunnels had a slightly smaller impact on climate change and that variants that were relatively uniform in phases showed the least financial and implementation risks.

Weighting the calculated transport criterium scores by 30%, risk criterium scores by 10%, and the other criteria scores by 12% (weight combination A, according to the ratios determined through a consultation process with stakeholders), resulted in ranking Variant 1 as the best, as shown in Table 6.

**Table 6.** Weighted criterium scores and a total variant score according to weight combination A.

| Criteria | Variant | | | | | |
|---|---|---|---|---|---|---|
| | 1 | 2 | 3 | 4 | 5 | 6 |
| Transport | 1.1 | 0.7 | 1.2 | 0.8 | 1.0 | 0.8 |
| Compliance | 0.4 | 0.4 | 0.4 | 0.4 | 0.3 | 0.3 |
| Cost | 0.5 | 0.3 | 0.3 | 0.3 | 0.5 | 0.3 |
| Environment | 0.4 | 0.3 | 0.2 | 0.3 | 0.4 | 0.5 |
| Institutional analysis | 0.4 | 0.4 | 0.4 | 0.3 | 0.3 | 0.2 |
| Implementation | 0.4 | 0.4 | 0.3 | 0.4 | 0.4 | 0.4 |
| Risk | 0.4 | 0.2 | 0.3 | 0.2 | 0.3 | 0.2 |
| **Total variant score** | **3.7** | **2.7** | **3.2** | **2.6** | **3.2** | **2.7** |

*3.4. Sensitivity Analysis*

The results of the sensitivity analysis performed over the calculated criterium scores are shown in Table 7.

**Table 7.** Total variant scores according to criteria weight combinations A to F.

| Criteria Weight Combinations | Variant | | | | | |
|---|---|---|---|---|---|---|
| | 1 | 2 | 3 | 4 | 5 | 6 |
| A | 3.7 | 2.7 | 3.2 | 2.6 | 3.2 | 2.7 |
| B | 3.7 | 3.0 | 3.1 | 2.7 | 3.0 | 2.6 |
| C | 3.9 | 2.8 | 2.9 | 2.5 | 3.4 | 2.6 |
| D | 3.7 | 2.7 | 2.8 | 2.6 | 3.3 | 2.9 |
| E | 3.7 | 2.9 | 3.1 | 2.6 | 3.1 | 2.4 |
| F | 3.7 | 2.9 | 3.0 | 2.7 | 3.2 | 2.8 |

The results show that the best-ranked variant was indeed Variant 1, regardless of the 30% weights shifts to criteria other than transport (weights B to F). This variant rank was also the least sensitive to applied weight shifts. On the other hand, the rank of Variant 6 was influenced the most by applied weight shifts, specifically when the highest weight of 30% was applied to the institutional analysis criterion (weight combination E).

## 4. Discussion

According to [44], methodological problems in the infrastructure evaluation process by MCA are (1) the definition and hierarchical order of project objectives, (2) the conflict between chosen criteria (e.g., minimal environmental consequences with maximum travel speed, or minimal construction costs), and (3) the weight of criteria. According to [47], the selection of objectives and criteria should be the result of global assessment (considering environmental and transportation policy, and spatial development concepts), as well as the assessment of local conditions that stem from the specificity of a given location. Therefore, in this investigation, the vision and objectives of the ZUA's transportation system development were defined based on the detailed traffic system functional analysis, the analysis of the European, national, regional, and local strategic documents' objectives,

spatial planning documentation on regional, city, municipal, and local scales, development masterplans, and projects.

According to [18], limitations in the application of MCA in the infrastructure evaluation are most pronounced when evaluating criteria that are entirely or predominantly qualitative in nature, which is based on insufficiently elaborated designs. Therefore, to minimize the uncertainty of the analysis results, as many as six variants of the railway route alignments for the resolution of the ZUA's transportation issues were developed on a preliminary design level. This made the quantification process of seven chosen criteria (by normalization of 32 indicator values) transparent and objective.

The performance of MCA supplemented with partial CBA resulted in assigned weighted scores for each variant, which allowed for the prioritization of variants and selection of the most preferable one. To avoid potential objections that the set of criteria weights used in the analysis was subjective and the result biased, the sensitivity analysis was carried out. This analysis increased the reliability of ranking and allowed the decision-makers to see the full spectrum of possible outcomes and select the most robust alternative, as predicted in [46].

It needs to be emphasized that the cost criterion value in this investigation was not obtained by a full-fledged CBA, which is customarily and methodologically conducted during feasibility studies. Indirect benefits have been considered through estimation in the transport criterion evaluation, and induced benefits, maintenance, and operation costs were omitted. However, this analysis has served the purposes of the preliminary feasibility study. Future investigations will be focused on defining parameters for additional traffic, spatial, and social indicators identified by [19], and on including a more detailed CBA in the evaluation model.

## 5. Conclusions

Today, the evaluation of planned rail infrastructure projects in the context of sustainable urban development is a fundamental requirement. Both the construction of new and the renewal of existing rail transportation facilities in urban areas has become increasingly important, considering the growing requirements for green mobility and connectivity. To choose an urban rail project solution that enables sustainable urban development, it is necessary to consider numerous quantitative and qualitative evaluation criteria: environmental, economic, engineering, social, and safety. Due to the sustainability criteria's specificity, and the premise that there are no "bad projects", but only wrong decisions when choosing the project evaluation procedure, the necessity to improve existing or create new complex rail infrastructure evaluation models that combine different methods is clear.

This paper presented the investigation of combining multi-criteria analysis with partial cost–benefit and sensitivity analyses for decision-making in the field of rail infrastructure in urban areas. The investigation was performed based on data obtained and created during the design of a pre-feasibility study for (re)integrating rail with road, maritime, and air transportation in the Zadar urban area. The investigation process was divided into four steps: (1) identification of vision, objectives, and measures for rail transportation system development, (2) design of six variants of railway route alignments, (3) evaluation of designed variants by performing multi-criteria analysis supplemented with partial cost–benefit analysis, and (4) sensitivity analysis of the evaluation results.

The results of the preliminary feasibility study showed that rail infrastructure revitalization, development, and integration present an opportunity to pursue a sustainable transportation system in the Zadar urban area by strengthening local public transportation and, consequently, reducing traffic congestion, increasing safety, and liberating space for environmentally friendly zero-emission mobility solutions. At the same time, the results provided the decision-makers with a sustainable approach for further elaboration of the spatial planning documents.

The results of the investigation showed that the use of multi-criteria analysis can contribute to the quality of decision-making in the field of urban rail infrastructure revi-

talization, development, and integration projects. The preconditions that must be met are well-defined project vision, objectives, and measures, the elaboration of the track design to the level of detail required to properly determine the indicator values of selected criteria, and a comprehensive selection of the criteria. If these are met, then the proposed model for project evaluation can ensure a high level of objectivity, auditability, and transparency of the urban rail infrastructure evaluation process, at least on the pre-feasibility study level.

**Author Contributions:** Conceptualization, M.A. and Ž.S.; methodology M.A.; formal analysis, M.A.; investigation, M.A., S.A., and I.M.; resources, Ž.S.; data curation, M.A., S.A., and I.M.; writing—original draft preparation, M.A. and S.A.; writing—review and editing, I.M. and Ž.S.; visualization, M.A. and I.M.; supervision, Ž.S.; project administration, Ž.S.; funding acquisition, Ž.S. All authors have read and agreed to the published version of the manuscript.

**Funding:** This research received no external funding.

**Data Availability Statement:** The data presented in this study are available on request from the corresponding author.

**Conflicts of Interest:** The authors declare no conflicts of interest. The funders had no role in the design of the study; in the collection, analyses, or interpretation of data; in the writing of the manuscript; or in the decision to publish the results.

## Appendix A

**Table A1.** PFS compliance to priorities and objectives of strategic documents at the European level.

| Document | Document Vision | PFS Complies to Priorities and Objectives |
|---|---|---|
| European Union's Strategic Agenda 2019–2024 [31] | Climate-neutral, green, fair, and social Europe has been built. European transportation infrastructure investments, planned in a way that maximized positive impact on economic growth and minimized negative impact on the environment, had a positive impact on economic growth, created wealth and jobs, and enhanced trade, geographical accessibility, and mobility. | Accelerating the transition to renewables and increasing energy efficiency. Investing in solutions for the mobility of the future. Improving the quality of our air and waters. |
| WHITE PAPER—Roadmap to a Single European Transportation Area [32] | European transportation infrastructure investments, planned in a way that maximized positive impact on economic growth and minimized negative impact on the environment, had a positive impact on economic growth, created wealth and jobs, and enhanced trade, geographical accessibility, and mobility. | Growing transportation and supporting mobility while reaching the 60% emission reduction target. Achieving clean urban transportation and commuting by a higher share of travel by collective transportation. Optimizing the performance of multimodal logistic chains, by greater use of more energy-efficient modes. |
| General Union Environment Action Program to 2030 [33] | Europeans live well, within planetary boundaries, in a well-being economy where nothing is wasted. Growth is regenerative, climate neutrality is a reality, and inequalities are significantly reduced. | Achieving the 2030 greenhouse gas emission reduction target and climate neutrality by 2050. Enhancing adaptive capacity, strengthening resilience, and reducing vulnerability to climate change. Advancing towards a regenerative growth model, decoupling economic growth from resource use and environmental degradation, and accelerating the transition to a circular economy. Pursuing a zero-pollution ambition, including for air, water, and soil, and protecting the health and well-being of Europeans. Protecting, preserving, and restoring biodiversity, and enhancing natural capital. Reducing environmental and climate pressures related to production and consumption (particularly in the areas of energy, industry, buildings and infrastructure, mobility, tourism, international trade, and the food system). |

**Table A2.** PFS compliance to priorities and objectives of strategic documents at the national level.

| Document | Document Vision | PFS Complies to Priorities and Objectives |
|---|---|---|
| National development strategy of the Republic of Croatia until 2030 [34] | Economic and social development in the Republic of Croatia is in balance with nature through encouraging competitiveness and innovation of the economy and society, recovery and strengthening resistance to crises, green and digital transition, and balanced regional development. | Introducing cleaner, cheaper, and healthier forms of transportation by promoting a safe and sustainable transportation policy. Strengthening the role of cities in the polycentric development of urban areas. |
| Spatial Development Strategy of the Republic of Croatia [35] | The development of spatial and traffic systems in the Republic of Croatia is planned at the local and regional levels. | Developing a sustainable economy and infrastructure systems. Using space sparingly and directing development activities towards already used land. Intensively developing rail, sea, river, and air transportation systems and improving the existing road networks. |
| Transportation Development Strategy of the Republic of Croatia 2017–2030 [36] | The development of transportation infrastructure in the Republic of Croatia enabled economic and social growth as well as international connectivity. Developed transportation infrastructure improved the regional exchange of goods and the accessibility to all economic, health, tourist, and other contents. | Changing the distribution of passenger traffic in favor of public transportation and forms of transportation with zero emission of harmful gases. Changing the distribution of cargo traffic in favor of rail and sea traffic and traffic by inland waterways. Developing the transportation system according to the principle of economic sustainability. Reducing the impact of the transportation system on climate change and the environment. Increasing the safety of the traffic system. Increasing the interoperability of the transportation system. Improving the integration of traffic modes. Further developing the Croatian part of the TEN-T network. |

**Table A3.** PFS compliance to priorities and objectives of strategic documents at the regional level.

| Document | Document Vision | PFS Complies to Priorities and Objectives |
|---|---|---|
| Traffic masterplan of the functional region North Dalmatia [29] | North Dalmatia is a functional region with high regional and local accessibility of regional centers achieved through balanced mobility developments. | Reducing the negative impacts of traffic on the environment. Improving the efficiency and sustainability of the transportation sector. Increasing the competitiveness of the economy. Improving the efficiency, safety, and protection of the transportation sector. Improving the level of management of the transportation system according to the principles of economic and social efficiency. Increasing the level of intermodality. Improving intercity, regional, and international passenger accessibility through the modernization of railway infrastructure. Improving the integration of maritime and rail transportation with other transportation modes for local and regional transportation (passenger and freight). |
| Zadar County Development Plan 2021–2027 [37] | Zadar county is a competitive and developed county that leads in the blue and green growth of the economy of Adriatic Croatia, and a county of innovative and sustainable development with a safe and stimulating environments for all its residents. | Ensuring safe and sustainable mobility. Improving transportation connections and modernizing transportation systems. Encouraging the development of intermodal transportation and establishing a multimodal transportation hub. Introducing a system of integrated passenger transportation. Encouraging the application of environmentally friendly transportation solutions. Increasing the level of efficiency and functionality of the transportation system in the tourist season and in the difficult weather conditions. |

**Table A4.** PFS compliance to priorities and objectives of strategic documents at the local level.

| Document | Document Vision | PFS Complies to Priorities and Objectives |
|---|---|---|
| Development strategy of the Zadar urban area 2014–2020 [24] | ZUA is integrally developed as a polyfunctional economy with sustainable use of spatial resources, increase in living standards, technological progress and innovations in the urban environment, and improved development management system. | Revitalizing rail freight and local passenger traffic (daily migration). Rebuilding and modernizing the Zadar–Knin railway for international mixed traffic. Intermodally connecting railway and seaports in Zadar. |

| Document | Document Vision | PFS Complies to Priorities and Objectives |
|---|---|---|
| Development strategy of the city of Zadar 2013–2020 [22] | Zadar is a developed, safe, and open city that continuously strives to create better living conditions for its population. The economic structure is characterized by the growth and sustainable development of the processing industry and tourism, which is profiled and strengthened in the off-season. Zadar is successfully integrated into international traffic flows, open to cooperation with other cities and regions. New projects changed the urban image of the city, making it unique, attractive, and pleasant for locals and numerous guests. | Ensuring the preservation and sustainable spatial development. Improving the infrastructure system. Ensuring the intermodality of traffic routes. Directing traffic from roads to railways and coastal and inland navigation. |
| Bibinje Municipality Total Development Program 2013–2018 [38] | Bibinje is a developed, safe, and open municipality that continuously strives to create better living conditions for its population with a special emphasis on caring for young people. The fundamental drivers of progress are small and medium enterprises, tourism, and agriculture. | Developing basic infrastructure. Revitalizing attractive coastal areas by dislocating the railway. |
| Sukošan Municipality Strategic Development Program 2015–2020 [39] | Sukošan is a municipality with a preserved natural environment that provides its residents with a quality of life through a balanced evaluation of diverse natural resources and cultural heritage. | Increasing traffic safety through the modernization of railroad-level crossings. |

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
