# Peer review of "Contribution to Rail System Revitalization, Development, and Integration Projects Evaluation: A Case Study of the Zadar Urban Area"

_infrastructures, doi:10.3390/infrastructures9020032_

Round 1

Reviewer 1 Report

Comments and Suggestions for Authors

1)This work is a general case study. The results should be highlighted with some statistical results in the abstract.

2) Railway infrastructure planning is a crucial point that should be addressed considering various factors. It is recommended to include some classical references, like [1-3] in different areas of railway engineering to give readers a full picture of the problem.

[1] Liu, Zhigang, et al. "Review of Perspectives on Pantograph-Catenary Interaction Research for High-Speed Railways Operating at 400 km/h and Above." IEEE Transactions on Transportation Electrification (2023).

[2] Song, Yang, et al. "A spatial coupling model to study dynamic performance of pantograph-catenary with vehicle-track excitation." Mechanical Systems and Signal Processing 151 (2021): 107336.

[3] Zhai, Wanming, et al. "Train–track–bridge dynamic interaction: a state-of-the-art review." Vehicle System Dynamics 57.7 (2019): 984-1027.

3) How to avoid a biased outcome by using the present method? Some necessary descriptions could be included in the final paragraph of the introduction.

4) A more specific title can be given for section 2. The current one ‘materials and methods’ is too general.

5) The railway line is not very distinct in Figure 1. It is recommended to replot the railway lines with a more distinct colour and wider line.

6) The optimisation algorithm can be described in more detail. What is the objective and what are the constraints?

Reviewer 2 Report

Comments and Suggestions for Authors

The article is of interest. The text of the article, as well as the research methodology, are presented in a very accessible manner.

But there are several issues that need to be taken into account.

1. The analysis of the literature sources given in section 1 must be strengthened with critical analysis. This will highlight a previously unresolved problem more clearly.

2. The article contains the numbering of Figure 3 twice. Apparently, the authors made a typo. This needs to be fixed.

3. It is not entirely clear how the risk criterion was calculated. I think the authors need to explain this more clearly in the text of the article.

4. In my opinion, it is better to show the “Discussion” and “Conclusions” sections separately.

5. In the Discussion section, it is necessary to emphasize the disadvantages and advantages of this study in comparison with already known results. It is also necessary to provide the limitations of this study and prospects for its further development.

Reviewer 3 Report

Comments and Suggestions for Authors

Dear authors,

thank you for an interesting read. Your views are quite impressive.

The article seems fine, but add these few comments:

- the article lacks a literature review section, it would be interesting to mention a few contributions that dealt with similar research and compare them with your idea.

- you could expand the discussion and conclusion section, or separate these two sections.

- figures 8,9,10 could be described more in the text.

Otherwise, the article has an interesting idea and a good presentation.

Thank you!

Reviewer 4 Report

Comments and Suggestions for Authors

The aim of the article was: Contribution to rail system revitalization, development, and integration projects evaluation - Zadar Urban Area Case Study

The article was written to a high standard and I do, however, have some observations and comments:

The abstract is of an appropriate length

Key words have been chosen appropriately

The literature review could be expanded with additional, more recent items.

Materials and Methods is adequately presented

Figures 1 and 6,7 lack a source citation

Figures 2 and 3 lack a description of the axes, I guess the year

Line 194 should be figure 5

Figure 2-5 does not contain the source of the data.

Figure 8-10 does not contain a description of the axes

Round 2

Reviewer 1 Report

Comments and Suggestions for Authors

Apparently, this paper does not implement a valid optimisation method to deal with the current problem. Only a simple sensitivity study is implemented. The scientific contribution is not sufficient for publication.
